# Crosstalk between Metabolite Production and Signaling Activity in Breast Cancer

**DOI:** 10.3390/ijms24087450

**Published:** 2023-04-18

**Authors:** Cankut Çubuk, Carlos Loucera, María Peña-Chilet, Joaquin Dopazo

**Affiliations:** 1Computational Medicine Platform, Andalusian Public Foundation Progress and Health-FPS, 41013 Sevilla, Spain; 2Centre for Experimental Medicine and Rheumatology, William Harvey Research Institute, Barts and The London School of Medicine and Dentistry, Queen Mary University of London, London E1 4NS, UK; 3Computational Systems Medicine, Institute of Biomedicine of Seville (IBiS), University Hospital Virgen del Rocío, Consejo Superior de Investigaciones Científicas, University of Seville, 41013 Sevilla, Spain; 4Centro de Investigación Biomédica en Red de Enfermedades Raras (CIBERER), 41013 Sevilla, Spain; 5FPS, ELIXIR-es, Hospital Virgen del Rocío, 42013 Sevilla, Spain

**Keywords:** metabolism, signaling pathway, crosstalk, machine learning, mathematical modeling, artificial intelligence, breast cancer

## Abstract

The reprogramming of metabolism is a recognized cancer hallmark. It is well known that different signaling pathways regulate and orchestrate this reprogramming that contributes to cancer initiation and development. However, recent evidence is accumulating, suggesting that several metabolites could play a relevant role in regulating signaling pathways. To assess the potential role of metabolites in the regulation of signaling pathways, both metabolic and signaling pathway activities of Breast invasive Carcinoma (BRCA) have been modeled using mechanistic models. Gaussian Processes, powerful machine learning methods, were used in combination with SHapley Additive exPlanations (SHAP), a recent methodology that conveys causality, to obtain potential causal relationships between the production of metabolites and the regulation of signaling pathways. A total of 317 metabolites were found to have a strong impact on signaling circuits. The results presented here point to the existence of a complex crosstalk between signaling and metabolic pathways more complex than previously was thought.

## 1. Introduction

The link between metabolism and cancer has been well known since almost one century ago [1], with the observation of enhanced aerobic glycolysis (also known as the Warburg effect) [2]. Actually, the common proliferative phenotype of cancer cells relies on the biosynthesis of cellular components and on the generation of energy, which are attained by reprogramming metabolism [2,3]. Besides the Warburg effect, other alterations in the synthesis of nucleotides, amino acids and lipids [4,5]; mutations in metabolic genes [6]; and accumulations of key metabolites [7] are also common in cancer. Consequently, reprogramming of cellular metabolism is an essential factor for cancer development and progression [8], and is, therefore, a recognized neoplastic hallmark [8,9]. Such changes are driven by modifications in the activity of key signaling pathways which are important regulators of metabolism [10]. It is well known that signaling controls the metabolism through different oncogenic pathways [11] such as the Hippo pathway which promotes glycolysis [12], the PI3K-AKT/mTOR Pathway that regulates diverse metabolic enzymes [13], and the Myc Pathway that regulates the expression of genes associated with glucose, glutamine and fatty acid metabolism [14], to cite just a few. Thus, in the conventional representation of causality, signaling precedes metabolism in cancer [10,11]. However, there is increasing evidence that different types of metabolites, traditionally associated with bioenergetics or biosynthesis, also play relevant role in non-metabolic signaling functions [15]. Nevertheless, this aspect has been scarcely explored yet.

The profusion of detailed biological knowledge on signaling and metabolism, as stored in pathway repositories such as KEGG [16], Reactome [17] or WikiPathways [18] has provided a conceptual framework for the development of accurate models of pathway activity, some of them involving the notion of causality. Conventional approaches such as Constraint-Based Models (CBMs), which use maps of metabolic networks in combination with gene activity inferred from transcriptomic profiles, have been applied to decipher relationships between various aspects of the cellular metabolism and phenotypes [19]. CBMs have been used for the analysis of metabolism in different scenarios [20]. Other strategies that use mechanistic models applied to metabolic pathways [21] have successfully been applied to predict gene essentiality in cancer [22]. Other modeling strategies have been used for the reconstruction of genome-scale metabolic networks [23]. In particular, mechanistic models of signaling pathways have successfully been used to deconvolute disease mechanisms behind different cancers [22,24] (including neuroblastoma [25,26] and glioblastoma [27]), the mechanisms of drugs action [28] or gender-specific effects of drugs in cancer [29]. It is worth noting that mechanistic models has been used to predict drugs that could be efficient COVID-19 treatments [30] and the predictions were validated using a cohort of almost 16,000 patients [31,32].

Here, a study on the crosstalk between metabolism and signaling has been carried out in BRCA, using mechanistic models to simultaneously infer from gene expression data the production of metabolites [21,22] and the activity of signaling pathways [24,33]. Importantly, the use of machine learning methods, such as Gaussian Processes (GP) [34] along with the SHapley Additive exPlanations (SHAP) method [35] that allows exploring causality, has allowed to relate the production of certain metabolites to the different activation statuses of several signaling circuits. The results obtained support a scenario of cross talking between signaling and metabolism more complex than previously thought.

## 2. Results

### 2.1. Differential Metabolite Production

Gene expression data for BRCA and controls were downloaded from the TCGA portal and processed as described in Materials and Methods Section 4. Metabolica [24], the extended version of the Metabolizer method, was applied in a cancer versus normal tissue comparison. A total of 149 metabolites showed a significant differential production rate. The complete list of differentially produced metabolites (DPMs) is provided in Appendix A. When compared with the differential abundance of metabolites, as reported by the metabolomic study [36], 31 (27%) of them were overlapping. This suggests that a considerable amount of metabolite production predicted by the Metabolica [21] algorithm is not transient and can be detected by experimental metabolic profiling.

### 2.2. Machine Learning Performance

As outlined in Methods, a 100-times repeated 5-fold cross-validation was carried out in order to measure the viability of the model in terms of the *R*^2^ score and mean squared error (*MSE*). As expected, not all tasks (circuit activities) can be predicted with reliability. However, there are no clear signs of overfitting, as can be seen by inspecting the joint distribution plot (Figure 1, left) of the mean task-wise *R*^2^ score over the train and test folds: there is a clear linear trend between the task-wise average score of each training split and its paired mean prediction over the test sets. Furthermore, there is a high-density area around x ≥ 0.5, y ≤ 0.5, and to be conservative, we propose a test *R*^2^ score above 0.5 as the threshold for annotating any given metabolite. On the other hand, the right-hand side of Figure 1 shows the *MSE*, averaged for each signaling pathway, across the training and test split iterations of the proposed cross-validation procedure. The 95% confidence intervals for the *MSE* (represented as bars) show that the variability decreases as the predictive performance increases (less is better), for both the test and train fold sets. As with the *R*^2^ score, the error trend is clearly linear and close to the diagonal (identity).

### 2.3. Cross-Talk between Metabolism and Signaling

Using the multi-task Gaussian process, a total of 317 metabolites were found to have a strong impact on the activity of signaling circuits in BRCA (Appendix A). These circuits were annotated by Uniprot functions and cancer hallmarks as described in Materials and Methods Section 4. The distribution of both annotations is depicted in Figure 2.

Since the activity of a number of signaling circuits can be predicted from the previous production of metabolites, it is likely that such metabolites are the ultimate trigger for this cross-talk. However, signaling and metabolic pathways share some genes (208 out of a total of 3349 genes in signaling circuits and 1419 genes in metabolic circuits, see Appendix A) and, consequently, part of this cross-talk could be due simply to the simultaneous activation of both circuits due to common genes. To distinguish between these two scenarios, the distribution of common genes across relevant and non-relevant circuits was studied. Thus, a Fisher exact test was used to detect if, when one metabolite predicts the behaviors of one signaling circuit, there is significant enrichment of common genes between the signaling circuit and the metabolic circuit corresponding to the metabolite. Then, a 2 × 2 contingency table was constructed for relevant (*R*^2^ score > 0.5) and non-relevant (*R*^2^ < 0.5) predictions that have or have not common genes. As a result, signaling circuits predicted and metabolic circuits corresponding to the most predictive metabolite do not show any significant enrichment in common genes (*p*-value = 0.99). Actually, if the circuits analyzed are expanded to those corresponding to the 20 best predictive metabolites, the scenario is the same (*p*-value = 1). This discards common genes between metabolic and signaling pathways as an explanation for this cross-talk, reinforcing the hypothesis that the metabolites detected are triggers of signaling pathway activity.

## 3. Discussion

The success of molecular targeted therapies in complex diseases relies on the complete view of the architecture of cellular systems and the role of components that construct this system of complex interactions. To date, mechanistic models of signaling pathways have demonstrated their usefulness in understanding the disease mechanisms behind different cancers [22,24] (including neuroblastoma [25,26] and glioblastoma [27]) cancer-prone rare diseases [37], the intricate mechanisms of action of drugs [28] or to deconvolute gender-specific effects of drugs in cancer [29]. Similarly, genome-scale models of cancer metabolism have also been used in cancer studies [38,39]. For example, Constraint-based flux Balance analysis Methods (CBM) [19] have been used for the characterization of oncometabolites [40,41]. Other mechanistic approaches, such as the Metabolizer [21], or its extended version used here, the Metabolica, focused on the changes in the integrity of sub-pathways that lead to the production of different metabolites, no matter these are transient and do not cumulate [21,22]. This allows detecting potential changes in metabolites that could play a relevant role and being not detected in conventional genome-scale modeling strategies and not even in metabolomic experiments.

As in other whole-genome metabolomics studies, an exhaustive description of all the findings is not feasible. Therefore, only relevant cancer-related metabolites will be commented here. For example, the production of Adenylosuccinate, an indicator of breast and other cancers [42] has been detected by Metabolica (*p*-value = 3.38 × 10^−15^) but could not be detected in the metabolomic profiles (Appendix A). Another example is the significantly different synthesis of 2-aminoadipate that is detected by this approach (*p*-value = 1.02 × 10^−62^), but not using an experimental metabolomics approach (Appendix A). This metabolite is a potential modulator of glucose homeostasis; moreover, it has been identified as a marker of tumor aggressiveness in glioblastoma [43] and is altered in TNBC cell lines after administration of PARP inhibitor Veliparib [44]. In fact, it has been demonstrated that lysine metabolism is a relevant process in cancer [45,46]. Other interesting findings are the differences in phenylalanine production (*p*-value = 8.36 × 10^−20^), a protective amino acid against cancer [47] or capric acid (*p*-value = 2.21 × 10^−12^), a non-competitive AMPA receptor antagonist, responsible for the mitochondrial proliferation, that has been associated with overall breast cancer risk [48]. Moreover, many DNA precursors were found, including adenine, 2′-deoxyguanosine, or hypoxanthine, probably due to the higher replicative requirements of tumor growth, that may be indicative of cell proliferation and, therefore, its production levels may be used as a surrogate marker of aggressiveness; interestingly, nucleotide synthesis is reemerging as a metabolic vulnerability in cancer [49]. Indeed, HPRT1, an enzyme that uses hypoxanthine metabolite to recycle purines (in order to be more efficient in DNA and RNA synthesis), predicts clinical outcome and controls gene expression in breast cancer [50]. Moreover, this mechanistic modeling method predicts a differential production of Inosine (*p*-value = 2.17 × 10^−11^), again not detected in the conventional metabolomics experiment. This sensitivity in detecting metabolite production, even in absence of accumulation, can constitute an interesting alternative approach to evaluate immunotherapy response, since it has recently been demonstrated that inosine promotes T-cell-mediated tumor-killing activity in vitro [51] and, moreover, modulates response to checkpoint inhibitor immunotherapy [52]. There are also some natural nucleotides whose analogous compounds are used as chemotherapeutic agents, such as 2′-deoxycytidine and its analogue Decitabine, typically used in myelodysplastic syndromes (MDS), including leukemia. These are probably a result of the high replication rates characteristic of tumor cells [53].

As stated above, a complete and systematic description of the metabolic results is beyond the scope of this manuscript; however, it is worth mentioning some of the most relevant results found. It is interesting to note how the dysregulation of metabolites in BRCA, such as succinate and kynurenine (Appendix A), that are taking a key role in the initiation of tumorigenesis and its progression were predicted correctly [54,55]. Furthermore, the production of 5,6-dihydrouracil (Appendix A), an intermediate product of breakdown process of uracil into beta-alanine which is required for epithelial-mesenchymal transition [56], was found significantly altered in BRCA (*p*-value = 0.003), while it was not detected in the metabolomic profiles. This catabolic process is also called pyrimidine degradation module and was found to be essential for cancer cell survival in some tumor types and experimentally validated by us [22]. Some byproducts of lipid peroxidation reactions, such as cholesterol and 7-beta-hydroxycholesterol, were also differentially present in BRCA tumors. However, these reactions are enhanced by smoking, and dietary intake of meat, eggs and animal fat, so the potential causality may be taken with caution [57]. Indeed, current evidence suggests that oxysterols play a role in many cancers, including breast cancer [58].

Several sugars are found among the most relevant metabolites, including: maltose, fructose-6-phosphate, ribulose, xylulose and xylose, among others. It has been established that cancer cells reprogram their glucose metabolism to overcome increased ROS (Reactive Oxygen Species) [59,60]. In fact, ROS induce glycolysis upregulation in cancer cells, in a phenomenon known as the Warburg effect. Other metabolites found (glycerol and glycerophosphoethanolamine) can also be the result of this shift towards glycolysis, an inefficient metabolic pathway for energy metabolism, and a manifestation of the aforementioned Warburg effect found in cancer cells. Some authors hypothesized that the synthesis of glycerol phosphate is activated and maintained under glucose and serum starvation situations [61], since cancer cells more readily use glycolysis, even when sufficient oxygen is available. This reliance on aerobic glycolysis, and, therefore, the Warburg effect, promotes tumorigenesis and malignancy progression. [62]. Moreover, many polyamines appear among the most relevant metabolites, the most common among them are putrescine, spermine and spermidine, polycationic alkylamines commonly found in all living cells, and playing an important role in cell growth, proliferation, differentiation, migration, gene regulation, synthesis of proteins and nucleic acids, maintaining chromatin structure, regulating ion channels, maintaining membrane stability and scavenging free radicals [63]. Their involvement in several processes of cell growth and maintenance makes it obvious to think that its deregulation will also play an important role in diseases such as cancer. Indeed, an increase in intracellular polyamine concentrations has been found in cancer cells, associated with tumorigenesis. According to our results, polyamine metabolism has been found often dysregulated in cancers [64].

As previously commented, the influence of signaling over metabolism in cancer development and progression is well known [10,11]. The results presented here throw light over the less known role that metabolism may play over signaling [15]. When evaluating the relevance of metabolites to individual circuits (see Appendix A), a special focus has been made in those relevant to BRCA, such as NF-KB, PPAR, ErbB, TNF, Estrogen signaling and others relevant to cancer in general, such as MAPK, Ras, Wnt, p53, PI3K-Akt, mTor signaling or Chemokines and inflammatory processes. For example, we have detected 1-methylnicotinamide (*p*-value 8.03 × 10^−7^, but also detected by metabolomics analysis), functionally; MNA (methylnicotinamide) induces T-cells to secrete the tumor-promoting cytokine tumor necrosis factor alpha, being an immune regulatory metabolite in ovarian cancer [65].

Another interesting pathway is fatty acid synthesis, which occurs in the cytoplasm, for which citrate (*p*-value 1.96 × 10^−8^) is the primary substrate; this pathway is activated in cancer and it is a metabolic hallmark of cancer cells. Indeed, it has been shown that extracellular citrate fuels cancer cell metabolism [66].

In addition to its canonical role as an amino acid for protein synthesis, asparagine has been found to have non-metabolic roles in regulating tumor-associated signaling [67]. Besides asparagine, there is a high representation of amino acids (alanine, arginine, tyrosine, glycine) within the list of differentially produced metabolites obtained with Metabolica, supporting the hypothesis that amino acids can facilitate the survival and proliferation of cancer cells under unfavorable situations, such as nutritional oxidative stress or starvation. As a matter of fact, targeting amino acid metabolism is becoming a potential therapeutic strategy for cancer patients [68].

Interestingly, Indolepyruvate (a byproduct of the tryptophan metabolism) is highly relevant to several circuits in NF-KB, TNF and chemokines signaling pathways, revealing a potential effect in BRCA tumors. Indolepiruvate has a role as a mechanism of innate immune evasion in some organisms, for example, the parasite Trypanosoma brucei produces metabolite indolepyruvate that decreases HIF-1α and glycolysis in macrophages [69]. Hence, tumor cells can be using this to avoid immune response. Indeed, the role of Trp metabolites and related enzymes in inflammation and cancer has been widely studied, showing a link to tumorigenesis [70] and to the establishment of an immunosuppressive microenvironment resulting in impaired immune response against tumor cells [71]. Besides Indolepyruvate, other Trp- related metabolites have shown relevant impact in BRCA pathways, such as 5-Hydroxy-N-formylkynurenine (Appendix A). It is also worth highlighting the impact of hippurate over PPAR signaling pathway in BRCA model (Appendix A), since circulating hippurate levels have been associated with pre-diagnosis risk to develop breast cancer in premenopausal women [72]; therefore, evaluating the production of hippurate can be a powerful tool in breast cancer risk management.

Overall, a relevant impact has been found for several vitamins and antioxidants, such as pantothenol, retinol, calciferol, tetrahydrofolate and carnitine (and derivatives), over general cancer-related circuits (Appendix A), supporting the association with cancer of those and other molecules with antioxidant and immune boosting effects [72,73,74]. However, further studies would be necessary to elucidate their specific role in cancer mechanisms.

Finally, it is interesting to note that this cross-talk between metabolism and signaling is likely to be mediated uniquely by the metabolites. The results document how a large number of common metabolites (almost 2/3) predict the activity of a large number of signaling circuits. Precisely, mechanistic modeling has already shown how different cancer types can show different strategies to activate or deactivate functionally related pathways [22,24,29], as reflected by the different molecular mechanisms behind the cross-talk between metabolism and signaling observed in BRCA. This suggests that metabolites could constitute potentially promising therapeutic targets for specific interventions.

## 4. Materials and Methods

### 4.1. Samples and Data Processing

RNA-seq counts and simple somatic mutations data for a total of 1072 samples, 959 corresponding to BRCA tumor and 113 normal breast tissue, were downloaded from The International Cancer Genome Consortium (ICGC) repository [75]. The trimmed mean of M-values (TMM) method [76] was used for gene expression normalization. Normalized samples were log-transformed and a truncation by quantile 0.99 was applied. The COMBAT method [77] was used for batch effect correction. Finally, the data were re-scaled between 0 and 1.

The ANNOVAR tool [78] (v2017Jul16 with ljb26 database) was used for functional characterization of non-synonymous genetic variants. The variants predicted as damaging by at least three out of five in silico pathogenicity predictors were considered loss-of-function (LoF) mutations. The in silico methods used were: SIFT [79], Polyphen2 [80], FATHMM [81], MutationTaster [82] and MutationAssessor [83]. For each tumor sample, expression value of the genes that were affected by the damaging variants were multiplicated by a decreasing constant: 0.001, to simulate the effect of LoF in the enzyme (equivalent to a non-expressed gene) [84,85].

Scaled and imputed metabolomics datasets of paired BRCA (65 tumor and 65 normal breast tissue) samples were downloaded from the Appendix A of a publication [36]. Additionally, quantile normalization was applied to these datasets using preprocessCore Bioconductor package [42].

### 4.2. Estimation of Signaling Pathway Activity

The Hipathia mechanistic model, which models the activity of signaling pathways from gene expression data [24], was used to estimate signaling activity. In particular, a R/Bioconductor implementation (v2.14.0) was used [33]. Hipathia uses KEGG [16], signaling pathways, which are decomposed into elementary signaling circuits, which can be considered self-regulating functional units of the cell [24,33,86]. In Hipathia, circuits are described as directed graphs that connect receptor proteins to effector proteins through a chain of activations and inhibitions exerted by intermediate proteins. The mathematical model estimates the transduction of the signal considering the level of activity of the protein nodes (using gene expression as proxies) that compose the circuit. In this way, Hipathia transforms gene expression measurements into signaling circuit activities, and, therefore, functional profiles of cell activity [24].

### 4.3. Estimation of Metabolite Production

An extended version of the Metabolizer (v1.7.0) algorithm [21,22], Metabolica, was used to estimate the potential production of metabolites from the measurement of the gene expression values corresponding to the enzymes involved in the reactions. Metabolica requires the definition of sub-pathways in which the activity of the reactions of metabolite synthesis are modeled. Breaking down a pathway into sub-pathways and estimating the activity of a sub-pathway do not depend on a particular pathway repository, but require essential information for metabolic reactions (substrate, product, and reversibility descriptions). Here, the canonical pathways presented in the KEGG database [16] were used. A total of 79 human metabolic pathways, containing 1901 reactions and 1270 metabolites (Appendix A), were downloaded. The KGML files were parsed using the KEGGgraph Bioconductor package [87] (v1.58.3). The sub-pathway that produces a given metabolite is defined by all the nodes which were visited inside its pathway using breadth-first search algorithm. This process starts from the metabolite produced (so-called product) and continues iteratively in the direction of the edges which are arriving at the product and its connected neighbor nodes. Figure 3 shows an example of a sub-pathway which is extracted from its pathway as described above.

Due to the highly interconnected nature of metabolic pathways and the numerous feedback loops, the convergence of calculations is challenging when the propagation algorithms are applied on metabolic hypergraphs. To deal with this issue, the feedback loops which are not derived from the product were kept; however, all the feedback loops (outdegree edges) of the product were removed. By this means, we also restrict the consumption of the product by its producing pathway.

Similar to the signaling [24] or metabolic module [21] implementations, Metabolica requires starting node(s) to initialize the propagation of metabolic flux along sub-pathways. The definition of starting nodes is a two-step process: first, the metabolites with indegree of zero and the metabolites at the farthest position in the sub-pathway (from the product) are selected, and the propagation algorithm (see below) is run without any objective function. In the second step, the nodes which were not visited in the previous run are included in the list of starting nodes in order to guarantee that all the nodes can be visited in the further runs. All the decomposing steps were performed only one time and saved for future analysis of the pathway.

For each reaction node *r**_i_*** of a sub-pathway, the metabolic flux propagation is computed by the given formula according to the recursive rules depicted in Figure 4.

*wm_s_,r_i_* is the amount of substrate (*m_s_*) used by reaction (*r_i_*), *wr_i,_m_p_* is the amount of product (*m_p_*) produced by the reaction ***r_i_*** and *m_p_* is the final amount of product which is produced by different reactions. *N*^+^ and *N*^−^ denote the neighborhood of a node in the direction of its outgoing and incoming edges, respectively. *n* is the total number of *wr_i,_m_p_* plus 1 for *m_p_*. Thus, the Equation (1) distributes the substrate proportionally with the activities of its consuming reactions. The Equation (2) aims to elucidate the reaction rate (limited by the minimum amount of the substrates used). The amount of metabolite produced per unit time depends on the capacity of enzyme (saturation) and the amount of substrate. This is the combination of Michaelis–Menten kinetics and systems-level analysis of mechanisms regulating metabolic fluxes [88]. The Equation (3) updates *m_p_* node with the amount of contributing product of the reaction *r_i_* without saturating this node and it can also handle the loops appropriately. The loops in a sub-pathway need a high number of iterations to stabilize the flux propagated. Thus, Metabolica iterates the flux that is in a loop until it reaches the convergence state. Here, the convergence state is defined as almost-zero flux change between iterations. Therefore, Metabolica repeats steps 1, 2 and 3 until the flux initiated in the initial nodes reaches the product in a sub-pathway and while the flux which is propagated in a loop has not reached convergence. Metabolica input values in the [0,1] interval and returns output values in the same interval. Such results are non-dimensional values that, like gene expression values, can be interpreted in the context of a class comparison.

### 4.4. Differential Activation and Metabolic Production Analysis

The mechanistic model implemented in Hipathia infers signaling circuit activities from the expression levels of the genes corresponding to the proteins involved in the circuits.

Similarly, Metabolica estimates the production activity of the metabolites using the reactions involved in the sub-pathways analyzed [21]. It is important to note that Metabolica does not necessarily account for metabolite accumulation in the cell, which can be detected in metabolomics experiments, but rather for metabolite production (no matter if it is further consumed in another reaction and its existence is transient). Important metabolites can have a transient presence in the cell but still play an important role in cancer.

Finally, a Student’s *t*-test for paired samples is used to assess the significance of observed changes in metabolite concentrations or in metabolic production activity when samples of two conditions are compared.

A False Discovery Rate (FDR) [89] is used to correct the effect of multiple testing in all the comparisons.

### 4.5. Machine Learning to Relate Metabolite Production to Signaling Activity

Relating the production of metabolites with the activity of signaling pathways involves solving several Multi-Task regression problems, one for each signaling pathway and tumor type (for machine learning tasks only tumor samples are used). Here, Gaussian Processes (GP) [34] were used to find metabolites that have an impact on circuits. To implement the final predictive model of a Sparse variational Multi-Task GP (SVGP), GPflow [90] (v 1.3.0), a scalable Gaussian Process library on top of TensorFlow v2.11 [91], was used. In broad terms, a SVGP constructs an approximation of the full GP model by selecting a small subset of inducing points that serve as the support, i.e., a sparse representation, which is learned by using a variational formula [92] that jointly infers both the support samples and the kernel hyperparameters, is built. Note that variational approximations are less prone to overfit the data [92], which is one of the major drawbacks of nonlinear learning methods, especially in the high dimensional low sample size scenario.

The model assumes that the outputs (*n_s_* signaling circuit activities) can be expressed as *n_l_* latent GPs of the inputs (*n_m_* metabolite production activities) mixed by *W* (a real valued matrix of dimensions *n_s_* × *n_l_*). The covariance function of the GP is built using a linear combination of the squared exponential (also known as radial basis function kernel) and linear covariance functions. The kernel hyperparameters are forced to be shared across all the inputs (as many latent GPs as output dimensions, *n_l_
*= *n_s_*, are used), the inducing variables are shared between all the inputs (*n_i_
*= 50 inducing points are used) and W is the identity matrix of dimension *n_l_*. Note that Sparse Variational Processes are known to correctly scale across the data and output dimensions and all the computations are possible due to the data generating processes (all outputs are observed for every input) [93].

To assess the quality of the model, a 100-times repeated 5-fold cross-validation schema was performed in order to measure the generalization performance of the model. *R*^2^ scores (also known as the coefficient of determination) and mean squared errors (*MSE*) were calculated for each circuit to evaluate the performance of each model. Note that since the model hyperparameters are optimized as part of the GP modeling, *R*^2^ is indeed measuring the suitability of the model building procedure which is in line with recent recommendations for machine learning use in computational biology [94]. Here the *R*^2^ ranges from -infinite to 1 where 0 is the baseline prediction (predicting the mean of each output). The *R*^2^ score for a given signaling circuit and its prediction is given by:$$R^{2}\left(s,{\hat{s}}_{i}^{j}\right)=1-\frac{\sum_{i=1}^{n}{\left(s,{\hat{s}}_{i}^{j}\right)}^{2}}{\sum_{i=1}^{n}{\left(s_{i}-mean\left(s^{j}\right)\right)}^{2}}$$
where *n* is the number of samples, *j* indexes the signaling circuit, *i* the samples, and ^ refers to the mean prediction of the gaussian process model. While the mean squared error (*MSE*) is given by:$$MSE\left(s^{j},{\hat{s}}^{j}\right)=\frac{1}{n}\sum_{i=0}^{n-1}{\left(s_{i}^{j}-{\hat{s}}_{i}^{j}\right)}^{2}$$

Shapley values are a game theory approach to estimate the importance of any individual player as a part of a collaborative team [95], which could be used to better distribute the outcome of any given team-based game. The attribution model complies with the following properties: accuracy (additivity), consistency (symmetry) and nonexistence (null effect). The SHAP (SHapley Additive exPlanations) method [35] reinterprets the Shapley values in the context of machine learning interpretability by estimating the contribution that each predicting feature (metabolite production activity) has over the model output (the activity of signaling circuits). In this context, Shapley values explain how the prediction of any given sample differs from the global average prediction, thus it allows for sample-wise explanations (which are additive by construction). By averaging the absolute value of these explanations across a set of samples a measure of the importance of any given metabolite is obtained [96].

Here, the kernel-SHAP method [35] (v0.24.0), a model-agnostic approach to approximate the feature contributions as Shapley values, is used. Due to the fact that the approximations require a background set of samples [35] the dataset is split into training (80%) and test (20%) sets. The GP model is fitted to the training samples, which are used as the background for the kernel-SHAP, whereas the remaining samples are used for the feature contribution estimations, avoiding the overconfidence which could happen if the SHAP computations were performed using the same samples as the background. Note that, since the GP model is vector-valued, one prediction for each output (circuit) is obtained. Thus, the explanations are task-specific (the differences are computed for each output). At the end, a ranked list of metabolites is obtained for each signaling circuit along with the *R*^2^ score of the training and testing splits.

### 4.6. Annotation of Circuit Activity with Cell Functionalities and Cancer Hallmarks

The functionality triggered by signaling circuits is assumed to correspond to the functionality of the effector protein [24,33,86]. The effector proteins, which are found at the end of each signaling circuit that trigger the cellular functions, were annotated by the Uniprot protein functions [97] and cancer hallmarks [8]. The Cancer Hallmarks Analytics Tool (CHAT) API was used for the hallmark annotation of the circuits via a text mining approach [98].

Since signaling circuits trigger specific functionalities in the cell, metabolites that potentially regulate these signaling circuits can be annotated with the functions corresponding to these circuits. Appendix A lists the functions that the different metabolites would trigger due to their potential regulation of the corresponding signaling circuits (see Appendix A).

## 5. Conclusions

The results presented here reinforce the idea that, beyond the well-known influence of signaling on the reprogramming of metabolism in cancer [10,11], a non-negligible effect of metabolism over signaling activity seems to occur [15], suggesting the existence of a cross-talk between metabolism and signaling of a bigger magnitude than previously thought. Given all the above, an integral approach to study cancer origin and development must consider simultaneously signaling and metabolism to effectively identify new processes relevant in tumorigenesis and in cancer prognosis and treatment.

## Figures and Tables

**Figure 1 ijms-24-07450-f001:**
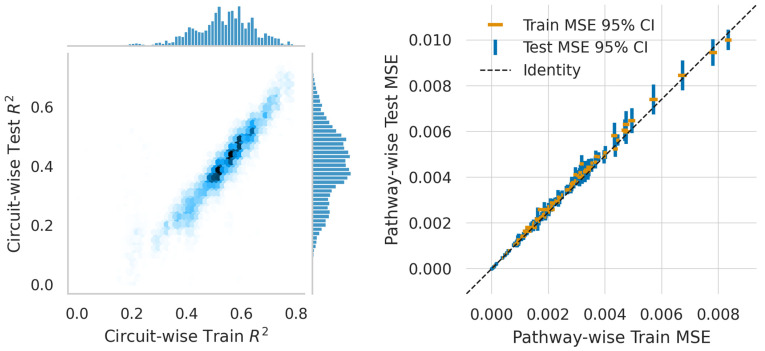
Machine learning method performance. Left: Joint distribution plot of the task-wise *R*^2^ for the train and test folds. Right: *MSE*, averaged for each signaling pathway.

**Figure 2 ijms-24-07450-f002:**
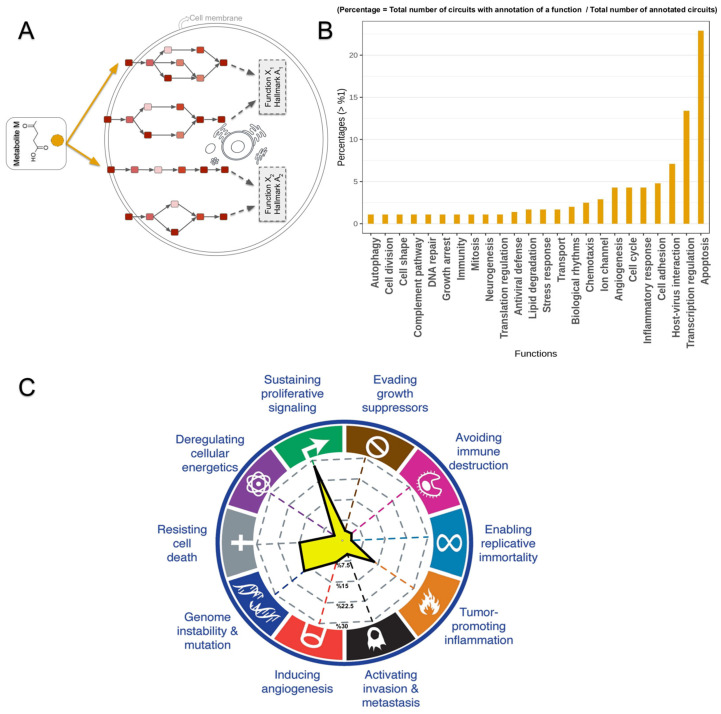
The profile of the annotated metabolites using the Multi-Task Gaussian Processes. (**A**) The illustration shows how cellular functions and hallmarks can be regulated by metabolites via different signaling circuits. A total of 317 metabolites displayed a relevant impact on signaling circuits. (**B**) The distribution of the percentages of the Uniprot functions corresponding to the functional activity of the signaling circuits over which the metabolites have a relevant impact. (**C**) Radar plot with the percentages of cancer hallmarks defined by the functional activity of the corresponding signaling circuits over which the metabolites have a relevant impact.

**Figure 3 ijms-24-07450-f003:**
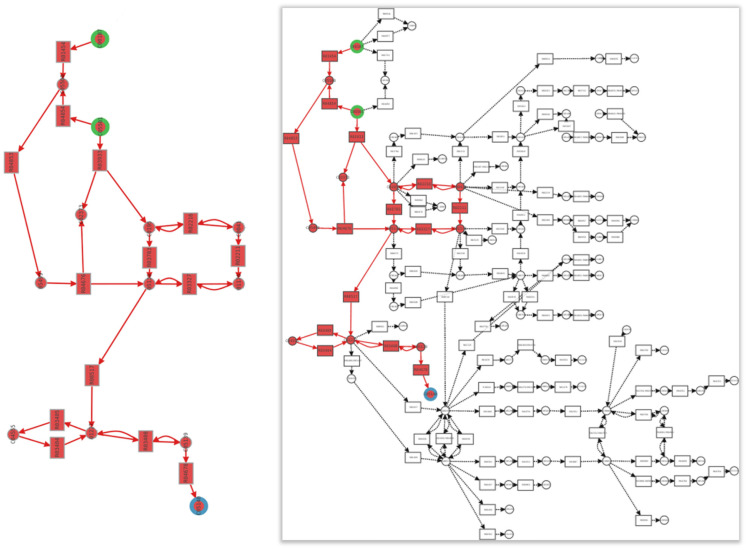
Example of a sub-pathway. The production sub-pathway of 4-Androsten-16alpha-ol-3,17-dione (blue node) starting from cholesterol and 20alpha,22beta-Dihydroxycholesterol (green nodes). This sub-pathway (**left**) was dissected from the steroid hormone biosynthesis pathway (**right**). The circles, rectangles and arrows are representing metabolites, metabolic reactions and reaction reversibility, respectively.

**Figure 4 ijms-24-07450-f004:**
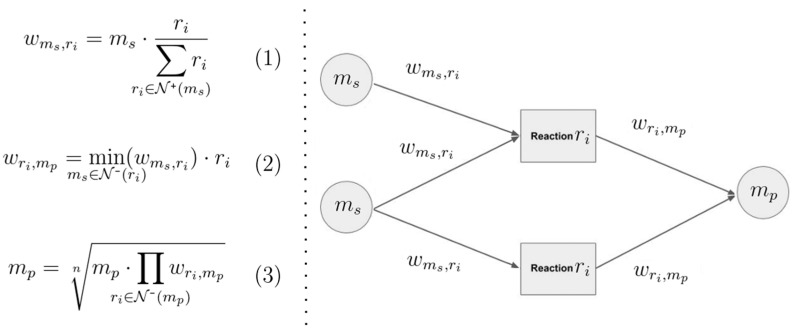
Recursive rules of the propagation algorithm.

## Data Availability

RNA-seq counts corresponding to BRCA samples were downloaded from The International Cancer Genome Consortium (ICGC) repository [75]. Scaled and imputed metabolomics datasets were downloaded from Supplementary Materials from [36]. Hipathia is freely available as an R/Bioconductor application at https://www.bioconductor.org/packages/release/bioc/html/hipathia.html (accessed on 5 March 2023), a web server http://hipathia.babelomics.org/ (accessed on 5 March 2023) and a cytoscape plugin https://apps.cytoscape.org/apps/cypathia (accessed on 5 March 2023) [33]. Metabolica is freely available at: https://github.com/babelomics/Metabolica (accessed on 5 March 2023).

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
