# Peer review of "Crosstalk between Metabolite Production and Signaling Activity in Breast Cancer"

_ijms, 2023, doi:10.3390/ijms24087450_

Round 1
Reviewer 1 Report
The authors propose a machine learning model (SHAP) to study the crosstalk between metabolism and signaling in a cohort of breast cancer. The study is comprehensive, and the result is validated using pathways analysis (KEGG and Reactome). The idea is novel and the manuscript is well-written. However, I have minor suggestions:
- The authors may add breast cancer to the keywords for better search retrieval.
-The authors may add recent genomic/metabolites works to the introduction such as PMID: 35205681 and PMID: 36768427.
- if the authors report the performance measurements for the machine learning method, it would be highly appreciated.
Author Response
COMMENT
=========
The authors propose a machine learning model (SHAP) to study the crosstalk between metabolism and signaling in a cohort of breast cancer. The study is comprehensive, and the result is validated using pathways analysis (KEGG and Reactome). The idea is novel and the manuscript is well-written. However, I have minor suggestions:
RESPONSE
========
We appreciate very much the comment of the referee
COMMENT
=========
- The authors may add breast cancer to the keywords for better search retrieval.
RESPONSE
========
Done
COMMENT
=========
-The authors may add recent genomic/metabolites works to the introduction such as PMID: 35205681 and PMID: 36768427.
RESPONSE
========
Thanks for the references. We have added them to the introduction
COMMENT
=========
- if the authors report the performance measurements for the machine learning method, it would be highly appreciated.
RESPONSE
========
We agree with the reviewer that an expanded discussion on the performance will improve the manuscript. To do so we have, on the one hand, updated Methods to include a better description of the R^2 score and the mean squared error metric. On the other hand, we have updated the performance discussion in the following way: i) we have unified the nomenclature when dealing with multiple outputs, using “pathway–wise” when summarizing the model performance across the outputs/circuits, and “circuit-wise” when the raw per-task/circuit metrics are used, ii) we have included a pathway-wise summarization of the cross-validation procedure in the form of a figure which includes 95% CI for the MSE, iii) we have updated the task/circuit-wise figure to reflect the changes (note there was also an error since we used an old version of the figure), and iv) the discussion on the performance has been expanded and updated to address the changes
Reviewer 2 Report
Comments to the Authors
This study uses computational methods to link metabolic signaling to signaling pathways. This is a very thought provocative hypothesis, which would revert common views.
Most of the emphasis lays on the method and less on the results, which are mostly presented in two figures (Figure 3 and 4). Figure 3 shows that the training and testing sets peak at R2=0.5. Figure 4b is the only panel with a real result, claiming 393 metabolites effecting signaling pathways. It is not clear to me which metabolites signal those pathways as in the supplemental tables have different numbers (S1: 9 uniques, S2: 149 uniques, S3 113 uniques). Please name those 393 metabolites.
On line 289 -299 numbers of genes are mentioned without naming them. Same on line 301 when 20 best predictive metabolites are mentioned but not named. Those need to be names together with their signaling pathways.
The first 3 paragraphs of the discussion (lines 306 – 358) does present some examples of metabolites modulating signaling pathways (this should go into the result section).
The authors write on line 349 that “a comprehensive description of the metabolic results is beyond the scope of this manuscript”. On the contrary, I actually think that is exactly what a reader expects is looking for in this study.
I propose that the authors expand their scope and present a more comprehensive analysis of the relevant metabolites effecting signaling pathways.
The authors should update their cancer hallmarks figure and statements as there are 14 hallmarks published recently by Hanahan in AACR’s Cancer Discovery (2022).
Author Response
COMMENT
=========
This study uses computational methods to link metabolic signaling to signaling pathways. This is a very thought provocative hypothesis, which would revert common views.
RESPONSE
========
We agree with the referee and appreciate the comment
COMMENT
=========
Most of the emphasis lays on the method and less on the results, which are mostly presented in two figures (Figure 3 and 4). Figure 3 shows that the training and testing sets peak at R2=0.5. Figure 4b is the only panel with a real result, claiming 393 metabolites effecting signaling pathways. It is not clear to me which metabolites signal those pathways as in the supplemental tables have different numbers (S1: 9 uniques, S2: 149 uniques, S3 113 uniques). Please name those 393 metabolites.
RESPONSE
========
Apologies, there has been a mistake. There are actually 317 metabolites. These metabolites have a relevant impact on signaling circuits. We have made the Supplementary tables more metabolite-centric for clarity.
COMMENT
=========
On line 289 -299 numbers of genes are mentioned without naming them. Same on line 301 when 20 best predictive metabolites are mentioned but not named. Those need to be names together with their signaling pathways.
RESPONSE
========
We have included a table with the names of the genes from signaling circuits, metabolic circuits and shared.
COMMENT
=========
The first 3 paragraphs of the discussion (lines 306 – 358) does present some examples of metabolites modulating signaling pathways (this should go into the result section).
RESPONSE
========
Actually, there are many metabolites that we mention in results in the tables. In the discussion we discuss the possible functional links of these metabolites (obviously not all of them), and this is why we mention some metabolites (as you request in your next comments).
COMMENT
=========
The authors write on line 349 that “a comprehensive description of the metabolic results is beyond the scope of this manuscript”. On the contrary, I actually think that is exactly what a reader expects is looking for in this study.
RESPONSE
========
We appreciate the interest of the referee on the biological side of the results and we agree that these are important. Following this suggestion, we have expanded the description of the metabolites found as relevant. However, we cannot expand the discussion to all the 317 metabolites. This is what we meant with the sentence, that we have rephrased.
COMMENT
=========
I propose that the authors expand their scope and present a more comprehensive analysis of the relevant metabolites effecting signaling pathways.
RESPONSE
========
Same as in the previous comment.
COMMENT
=========
The authors should update their cancer hallmarks figure and statements as there are 14 hallmarks published recently by Hanahan in AACR’s Cancer Discovery (2022).
RESPONSE
========
The assignment of circuits to cancer hallmarks was performed with the CHAT tool. Unfortunately, this tool has not been updated by the authors with the new hallmarks. We have thought of different ways to circumvent this problem, such as trying to do a manual assignment of functions, but this would entail the introduction of some bias in the new hallmarks. Since the classic hallmarks are still sufficiently informative and are not essential in the manuscript, as they have been used to represent and summarize the results, we have concluded that it is probably better not to try to include the new hallmarks, with the possible risk of bias that this entails.
Reviewer 3 Report
see the report.

Author Response
COMMENT
=========
The paper addresses the study of metabolites production in breast cancer signaling pathways activity.
The convential causality for signaling which precedes metabolism in cancer is proved to be not resilient in the general case, characterized according to a much more complex scenario.
A well posed introduction of the adopted propagation algorithm is followed by a detailed analysis of the statistical tools exploited, as Shapley values involved in the package used to achieve explainability. The promising results focus on the identification of metabolites characterized by a significant differential production rate, reinforcing the hypothesis that the metabolites detected are triggers of signaling pathway activity.
The paper might be suitable for publication in International Journal of Molecular Sciences.
RESPONSE
========
We appreciate the comments of the referee.
COMMENT
=========
However, some clarifications are necessary and some issues must be addressed.
- in lines 149 and 153 the notation wmsri and wrimp has to match the one used in Figure 2, so you have to correctly use pedices and avoid the capital letter for W;
RESPONSE
========
Apologies, we missed a comma in the notation. Also, we have used now a bigger lowercase “w”. We cannot use three levels of subindexes in the text.
COMMENT
=========
- in Figure 2 arrows from substrate to reaction node have to be visible;
RESPONSE
========
True!, we apologize for this oversight. We have changed the figure for a new one with the arrows visible.
COMMENT
=========
- in Section 2.5 in lines 217-228, the introduction of Shapley values should be supported by the following reference
ˆ Analyzing breast cancer invasive disease event classification through explainable artificial intelligence, Frontiers in medicine (2023),
where eXplainable Artficial Intelligence tools are used with heterogeneous clinical data of breast cancer patients.
RESPONSE
========
Done. Thanks for the reference. We were unaware of it.
Round 2
Reviewer 2 Report
The authors made satisfactory changes to the manuscript in response my comments.